# DETECTING PRETRAINING DATA FROM LARGE LANGUAGE MODELS

**Weijia Shi**[1] *   **Anirudh Ajith**[2]*   **Mengzhou Xia**[2]   **Yangsibo Huang**[2]
**Daogao Liu**[1]   **Terra Blevins**[1]   **Danqi Chen**[2]   **Luke Zettlemoyer**[1]
[1]University of Washington    [2]Princeton University
[swj0419.github.io/detect-pretrain.github.io](swj0419.github.io/detect-pretrain.github.io)

## ABSTRACT

Although large language models (LLMs) are widely deployed, the data used to train them is rarely disclosed. Given the incredible scale of this data, up to trillions of tokens, it is all but certain that it includes potentially problematic text such as copyrighted materials, personally identifiable information, and test data for widely reported reference benchmarks. However, we currently have no way to know which data of these types is included or in what proportions. In this paper, we study the pretraining data detection problem: *given a piece of text and black-box access to an LLM without knowing the pretraining data, can we determine if the model was trained on the provided text?* To facilitate this study, we introduce a dynamic benchmark WIKIMIA that uses data created before and after model training to support gold truth detection. We also introduce a new detection method MIN-K% PROB based on a simple hypothesis: an unseen example is likely to contain a few outlier words with low probabilities under the LLM, while a seen example is less likely to have words with such low probabilities. MIN-K% PROB can be applied without any knowledge about the pretraining corpus or any additional training, departing from previous detection methods that require training a reference model on data that is similar to the pretraining data. Moreover, our experiments demonstrate that MIN-K% PROB achieves a 7.4% improvement on WIKIMIA over these previous methods. We apply MIN-K% PROB to three real-world scenarios, copyrighted book detection, contaminated downstream example detection and privacy auditing of machine unlearning, and find it a consistently effective solution.

## 1    INTRODUCTION

As the scale of language model (LM) training corpora has grown, model developers (e.g, GPT-4 (Brown et al., 2020) and LLaMA 2 (Touvron et al., 2023b)) have become reluctant to disclose the full composition or sources of their data. This lack of transparency poses critical challenges to scientific model evaluation and ethical deployment. Critical private information may be exposed during pretraining; previous work showed that LLMs generated excerpts from copyrighted books (Chang et al., 2023) and personal emails (Mozes et al., 2023), potentially infringing upon the legal rights of original content creators and violating their privacy. Additionally, Sainz et al. (2023); Magar & Schwartz (2022); Narayanan (2023) showed that the pretraining corpus may inadvertently include benchmark evaluation data, making it difficult to assess the effectiveness of these models.

In this paper, we study the pretraining data detection problem: given a piece of text and black-box access to an LLM with no knowledge of its pretraining data, can we determine if the model was pretrained on the text? We present a benchmark, WIKIMIA, and an approach, MIN-K% PROB, for pretraining data detection. This problem is an instance of Membership Inference Attacks (MIAs), which was initially proposed by Shokri et al. (2016). Recent work has studied *fine-tuning* data detection (Song & Shmatikov, 2019; Shejwalkar et al., 2021; Mahloujifar et al., 2021) as an MIA problem. However, adopting these methods to detect the pertaining data of contemporary large LLMs presents two unique technical challenges: First, unlike fine-tuning which usually runs for multiple epochs, pretraining uses a much larger dataset but exposes each instance only once, significantly

---
*Equal contribution

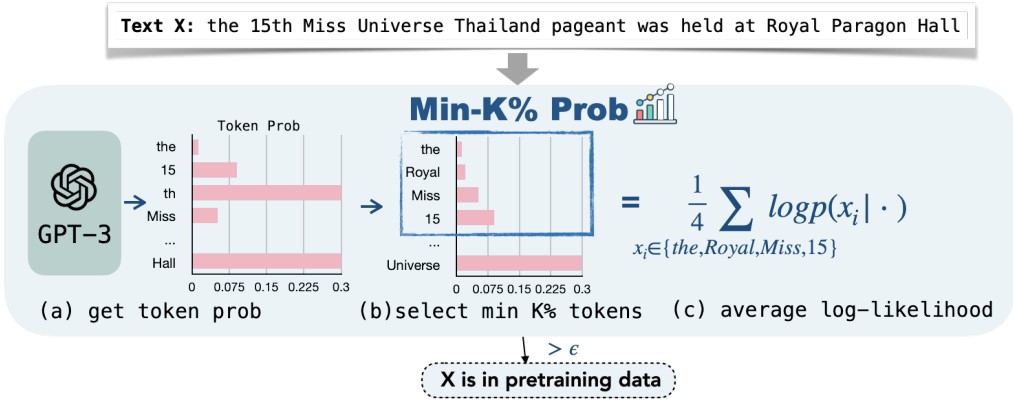

Figure 1: **Overview of MIN-K% PROB**. To determine whether a text $X$ is in the pretraining data of a LLM such as GPT, MIN-K% PROB first gets the probability for each token in $X$, selects the $k\%$ tokens with minimum probabilities and calculates their average log likelihood. If the average log likelihood is high, the text is likely in the pretraining data.

reducing the potential memorization required for successful MIAs (Leino & Fredrikson, 2020; Kandpal et al., 2022). Besides, previous methods often rely on one or more reference models (Carlini et al., 2022; Watson et al., 2022) trained in the same manner as the target model (e.g., on the shadow data sampled from the same underlying pretraining data distribution) to achieve precise detection. This is not possible for large language models, as the training distribution is usually not available and training would be too expensive.

Our first step towards addressing these challenges is to establish a reliable benchmark. We introduce WIKIMIA, a dynamic benchmark designed to periodically and automatically evaluate detection methods on any newly released pretrained LLMs. By leveraging the Wikipedia data timestamp and the model release date, we select old Wikipedia event data as our member data (i.e, *seen* data during pretraining) and recent Wikipedia event data (e.g., after 2023) as our non-member data (*unseen*). Our datasets thus exhibit three desirable properties: (1) **Accurate**: events that occur after LLM pretraining are guaranteed not to be present in the pretraining data. The temporal nature of events ensures that non-member data is indeed unseen and not mentioned in the pretraining data. (2) **General**: our benchmark is not confined to any specific model and can be applied to various models pretrained using Wikipedia (e.g., OPT, LLaMA, GPT-Neo) since Wikipedia is a commonly used pretraining data source. (3) **Dynamic**: we will continually update our benchmark by gathering newer non-member data (i.e., more recent events) from Wikipedia since our data construction pipeline is fully automated.

MIA methods for finetuning (Carlini et al., 2022; Watson et al., 2022) usually calibrate the target model probabilities of an example using a shadow reference model that is trained on a similar data distribution. However, these approaches are impractical for pretraining data detection due to the black-box nature of pretraining data and its high computational cost. Therefore, we propose a reference-free MIA method MIN-K% PROB. Our method is based on a simple hypothesis: an unseen example tends to contain a few outlier words with low probabilities, whereas a seen example is less likely to contain words with such low probabilities. MIN-K% PROB computes the average probabilities of outlier tokens. MIN-K% PROB can be applied without any knowledge about the pretrainig corpus or any additional training, departing from existing MIA methods, which rely on shadow reference models (Mattern et al., 2023; Carlini et al., 2021). Our experiments demonstrate that MIN-K% PROB outperforms the existing strongest baseline by 7.4% in AUC score on WIKIMIA. Further analysis suggests that the detection performance correlates positively with the *model size* and *detecting text length*.

To verify the applicability of our proposed method in real-world settings, we perform three case studies: copyrighted book detection (§5), privacy auditing of LLMs (§**??**) and dataset contamination detection (§6). We find that MIN-K% PROB significantly outperforms baseline methods in both scenarios. From our experiments on copyrighted book detection, we see strong evidence that GPT-3 [1] is pretrained on copyrighted books from the Books3 dataset (Gao et al., 2020; Min et al., 2023). From our experiments on privacy auditing of machine unlearning, we use MIN-K% PROB

---

[1]`text-davinci-003`.

to audit an unlearned LLM that is trained to forget copyrighted books using machine unlearning techniques (Eldan & Russinovich, 2023) and find such model could still output related copyrighted content. Furthermore, our controlled study on dataset contamination detection sheds light on the impact of pretraining design choices on detection difficulty; we find detection becomes harder when training data sizes increase, and occurrence frequency of the detecting example and learning rates decreases.

## 2 PRETRAINING DATA DETECTION PROBLEM

We study pretraining data detection, the problem of detecting whether a piece of text is part of the training data. First, we formally define the problem and describe its unique challenges that are not present in prior finetuning data detection studies (§2.1). We then curate WIKIMIA, the first benchmark for evaluating methods of pretraining data detection (§2.2).

### 2.1 PROBLEM DEFINITION AND CHALLENGES

We follow the standard definition of the membership inference attack (MIA) by Shokri et al. (2016); Mattern et al. (2023). Given a language model $f_\theta$ and its associated pretraining data $\mathcal{D} = \{z_i\}_{i \in [n]}$ sampled from an underlying distribution $\mathbb{D}$, the task objective is to learn a detector $h$ that can infer the membership of an arbitrary data point $x$: $h(x, f_\theta) \to \{0, 1\}$. We follow the standard setup of MIA, assuming that the detector has access to the LM only as a black box, and can compute token probabilities for any data point $x$.

**Challenge 1: Unavailability of the pretraining data distribution.** Existing state-of-art MIA methods for data detection during finetuning (Long et al., 2018; Watson et al., 2022; Mireshghallah et al., 2022a) typically use reference models $g_\gamma$ to compute the background difficulty of the data point and to calibrate the output probability of the target language model : $h(x, f_\theta, g_\gamma) \to \{0, 1\}$. Such reference models usually share the same model architecture as $f_\theta$ and are trained on shadow data $D_{\text{shadow}} \subset \mathbb{D}$ (Carlini et al., 2022; Watson et al., 2022), which are sampled from the same underlying distribution $\mathbb{D}$. These approaches assume that the detector can access (1) the distribution of the target model's training data, and (2) a sufficient number of samples from $\mathbb{D}$ to train a calibration model.

However, this assumption of accessing the distribution of pretraining training data is not realistic because such information is not always available (e.g., not released by model developers (Touvron et al., 2023b; OpenAI, 2023)). Even if access were possible, pretraining a reference model on it would be extremely computationally expensive given the incredible scale of pretraining data. In summary, the pretraining data detection problem aligns with the MIA definition but includes an assumption that the detector has no access to pretraining data distribution $\mathbb{D}$.

**Challenge 2: Detection difficulty.** Pretraining and finetuning differ significantly in the amount of data and compute used, as well as in optimization setups like training epochs and learning rate schedules. These factors significantly impact detection difficulty. One might intuitively deduce that detection becomes harder when dataset sizes increase, and the training epochs and learning rates decrease. We briefly describe some theoretical evidence that inform these intuitions in the following and show empirical results that support these hypotheses in §6.

To illustrate, given an example $z \in D$, we denote the model output as $f_\theta(z)$ Now, take another example $y$ sampled from $\mathbb{D} \setminus D$ (not part of the pretraining data). Determining whether an example $x$ was part of the training set becomes challenging if the outputs $f_\theta(z)$ and $f_\theta(y)$ are similar. The degree of similarity between $f_\theta(z)$ and $f_\theta(y)$ can be quantified using the total variation distance. According to previous research (Hardt et al., 2016; Bassily et al., 2020), the bound on this total variation distance between $f_\theta(z)$ and $f_\theta(y)$ is directly proportional to the *occurrence frequency of the example $x$*, *learning rates*, and the *inverse of dataset size*, which implies the detection difficulty correlates with these factors as well.

## 2.2 WikiMIA: A Dynamic Evaluation Benchmark

We construct our benchmark by using events added to Wikipedia after specific dates, treating them as non-member data since they are guaranteed not to be present in the pretraining data, which is the key idea behind our benchmark.

**Data construction.** We collect recent event pages from Wikipedia. **Step 1:** We set January 1, 2023 as the cutoff date, considering events occurring post-2023 as recent events (non-member data). We used the Wikipedia API to automatically retrieve articles and applied two filtering criteria: (1) the articles must belong to the event category, and (2) the page must be created post 2023. **Step 2:** For member data, we collected articles created before 2017 because many pretrained models, e.g., LLaMA, GPT-NeoX and OPT, were released after 2017 and incorporate Wikipedia dumps into their pretraining data. **Step 3:** Additionally, we filtered out Wikipedia pages lacking meaningful text, such as pages titled "Timeline of ..." or "List of ...". Given the limited number of events post-2023, we ultimately collected 394 recent events as our non-member data, and we randomly selected 394 events from pre-2016 Wikipedia pages as our member data. The data construction pipeline is automated, allowing for the curation of new non-member data for future cutoff dates.

**Benchmark setting.** In practice, LM users may need to detect texts that are paraphrased and edited, as well. Previous studies employing MIA have exclusively focused on detecting examples that exactly match the data used during pretraining. It remains an open question whether MIA methods can be employed to identify paraphrased examples that convey the same meaning as the original. In addition to the verbatim setting (*original*), we therefore introduce a *paraphrase setting* we leverage ChatGPT[2] to paraphrase the examples and subsequently assess if the MIA metric can effectively identify semantically equivalent examples.

Moreover, previous MIA evaluations usually mix different-length data in evaluation and report a single performance metric. However, our results reveal that data length significantly impacts the difficulty of detection. Intuitively, shorter sentences are harder to detect. Consequently, different data length buckets may lead to varying rankings of MIA methods. To investigate this further, we propose a *different-length setting*: we truncate the Wikipedia event data into different lengths—32, 64, 128, 256—and separately report the MIA methods' performance for each length segment. We describe the desirable properties in Appendix B.

## 3 Min-k% Prob: A Simple Reference-free Pretraining Data Detection Method

We introduce a pretraining data detection method Min-k% Prob that leverages minimum token probabilities of a text for detection. Min-k% Prob is based on the hypothesis that a non-member example is more likely to include a few outlier words with high negative log-likelihood (or low probability), while a member example is less likely to include words with high negative log-likelihood.

Consider a sequence of tokens in a sentence, denoted as $x = x_1, x_2, ..., x_N$, the log-likelihood of a token, $x_i$, given its preceding tokens is calculated as $\log p(x_i|x_1, ..., x_{i-1})$. We then select the $k\%$ of tokens from $x$ with the minimum token probability to form a set, Min-K%($x$), and compute the average log-likelihood of the tokens in this set:

$$\text{Min-k\% Prob}(x) = \frac{1}{E} \sum_{x_i \in \text{Min-K\%}(x)} \log p(x_i|x_1, ..., x_{i-1}). \tag{1}$$

where $E$ is the size of the Min-K%($x$) set. We can detect if a piece of text was included in pretraining data simply by thresholding this Min-k% Prob result. We summarize our method in Algorithm 1 in Appendix B.

---

[2]OpenAI. `https://chat.openai.com/chat`

## 4 EXPERIMENTS

We evaluate the performance of MIN-K% PROB and baseline detection methods against LMs such as LLaMA Touvron et al. (2023a), GPT-Neo (Black et al., 2022), and Pythia (Biderman et al., 2023) on WIKIMIA.

### 4.1 DATASETS AND METRICS

Our experiments use WIKIMIA of different lengths (32, 64, 128, 256), *original* and *paraphrase* settings. Following (Carlini et al., 2022; Mireshghallah et al., 2022a), we evaluate the effectiveness of a detection method using the True Positive Rate (TPR) and its False Positive Rate (FPR). We plot the ROC curve to measure the trade-off between the TPR and FPR and report the AUC score (the area under ROC curve) and TPR at low FPRs (TPR@5%FPR) as our metrics.

### 4.2 BASELINE DETECTION METHODS

We take existing reference-based and reference-free MIA methods as our baseline methods and evaluate their performance on WIKIMIA. These methods only consider sentence-level probability. Specifically, we use the *LOSS Attack* method (Yeom et al., 2018a), which predicts the membership of an example based on the loss of the target model when fed the example as input. In the context of LMs, this loss corresponds to perplexity of the example (***PPL***). Another method we consider is the neighborhood attack (Mattern et al., 2023), which leverages probability curvature to detect membership (***Neighbor***). This approach is identical to the DetectGPT (Mitchell et al., 2023) method recently proposed for classifying machine-generated vs. human-written text. Finally, we compare with membership inference methods proposed in (Carlini et al., 2021), including comparing the example perplexity to zlib compression entropy (***Zlib***), to the lowercased example perplexity (***Lowercase***) and to example perplexity under a smaller model pretrained on the same data (***Smaller Ref***). For the smaller reference model setting, we employ LLaMA-7B as the smaller model for LLaMA-65B and LLaMA-30B, GPT-Neo-125M for GPT-NeoX-20B, OPT-350M for OPT-66B and Pythia-70M for Pythia-2.8B.

### 4.3 IMPLEMENTATION AND RESULTS

**Implementation details.** The key hyperparameter of MIN-K% PROB is the percentage of tokens with the highest negative log-likelihood we select to form the *top-k%* set. We performed a small sweep over 10, 20, 30, 40, 50 on a held-out validation set using the LLAMA-60B model and found that $k = 20$ works best. We use this value for all experiments without further tuning. As we report the AUC score as our metric, we don't need to determine the threshold $\epsilon$.

**Main results.** We compare MIN-K% PROB and baseline methods in Table 1. Our experiments show that MIN-K% PROB consistently outperforms all baseline methods across diverse target language models, both in original and paraphrase settings. MIN-K% PROB achieves an AUC score of 0.72 on average, marking a 7.4% improvement over the best baseline method (i.e., PPL). Among the baselines, the simple LOSS Attack (PPL) outperforms the others. This demonstrates the effectiveness and generalizability of MIN-K% PROB in detecting pretraining data from various LMs. Further results such as TPR@5%FPR can be found in Appendix A, which shows a trend similar to Table 5.

### 4.4 ANALYSIS

We further delve into the factors influencing detection difficulty, focusing on two aspects: (1) the size of the target model, and (2) the length of the text.

**Model size.** We evaluate the performance of reference-free methods on detecting pretraining 128-length texts from different-sized LLaMA models (7, 13, 30, 65B). Figure 2a demonstrates a noticeable trend: the AUC score of the methods rises with increasing model size. This is likely because larger models have more parameters and thus are more likely to memorize the pretraining data.

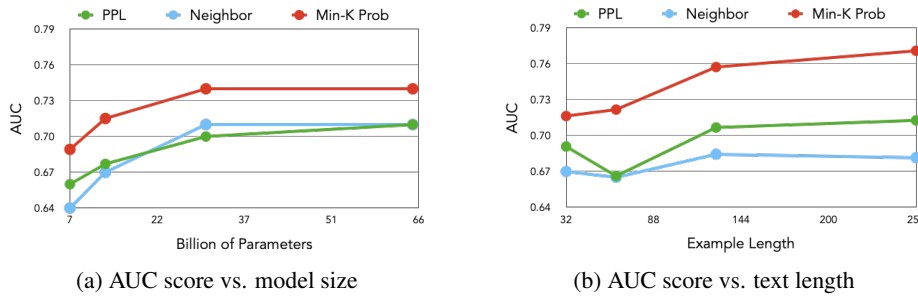

(a) AUC score vs. model size

(b) AUC score vs. text length

Figure 2: As model size or text length increases, detection becomes easier.

**Length of text.** In another experiment, we evaluate the detection method performance on examples of varying lengths in the original setting. As shown in Figure 2b, the AUC score of different methods increases as text length increases, likely because longer texts contain more information memorized by the target model, making them more distinguishable from the unseen texts.

Table 1: AUC score for detecting pretraining examples from the given model on WIKIMIA for MIN-K% PROB and baselines. *Ori.* and *Para.* denote the original and paraphrase settings, respectively. **Bold** shows the best AUC within each column.

| Method | Pythia-2.8B | | NeoX-20B | | LLaMA-30B | | LLaMA-65B | | OPT-66B | | |
|---|---|---|---|---|---|---|---|---|---|---|---|
| | *Ori.* | *Para.* | *Ori.* | *Para.* | *Ori.* | *Para.* | *Ori.* | *Para.* | *Ori.* | *Para.* | **Avg.** |
| Neighbor | 0.61 | 0.59 | 0.68 | 0.58 | 0.71 | 0.62 | 0.71 | 0.69 | 0.65 | 0.62 | 0.65 |
| PPL | 0.61 | 0.61 | 0.70 | 0.70 | 0.70 | 0.70 | 0.71 | 0.72 | 0.66 | 0.64 | 0.67 |
| Zlib | 0.65 | 0.54 | 0.72 | 0.62 | 0.72 | 0.64 | 0.72 | 0.66 | 0.67 | 0.57 | 0.65 |
| Lowercase | 0.59 | 0.60 | 0.68 | 0.67 | 0.59 | 0.54 | 0.63 | 0.60 | 0.59 | 0.58 | 0.61 |
| Smaller Ref | 0.60 | 0.58 | 0.68 | 0.65 | 0.72 | 0.64 | 0.74 | 0.70 | 0.67 | 0.64 | 0.66 |
| MIN-K% PROB | **0.67** | **0.66** | **0.76** | **0.74** | **0.74** | **0.73** | **0.74** | **0.74** | **0.71** | **0.69** | **0.72** |

In the following two sections, we apply MIN-K% PROB to real-world scenarios to detect copyrighted books and contaminated downstream tasks within LLMs.

## 5 CASE STUDY: DETECTING COPYRIGHTED BOOKS IN PRETRAINING DATA

MIN-K% PROB can also detect potential copyright infringement in training data, as we show in this section. Specifically, we use MIN-K% PROB to detect excerpts from copyrighted books in the Books3 subset of the Pile dataset (Gao et al., 2020) that may have been included in the GPT-3[3] training data.

### 5.1 EXPERIMENTAL SETUP

**Validation data to determine detection threshold.** We construct a validation set using 50 books known to be memorized by ChatGPT, likely indicating their presence in its training data (Chang et al., 2023), as positive examples. For negative examples, we collected 50 new books with first editions in 2023 that could not have been in the training data. From each book, we randomly extract 100 snippets of 512 words, creating a balanced validation set of 10,000 examples. We determine the optimal classification threshold with MIN-K% PROB by maximizing detection accuracy on this set.

**Test data and metrics.** We randomly select 100 books from the Books3 corpus that are known to contain copyrighted contents (Min et al., 2023). From each book, we extract 100 random 512-word snippets, creating a test set of 10,000 excerpts. We apply the threshold to decide if these books snippets have been trained with GPT-3. We then report the percentage of these snippets in each book (i.e., contamination rate) that are identified as being part of the pre-training data.

---

[3]text-davinci-003

## 5.2 RESULTS

Figure 3 shows MIN-K% PROB achieves an AUC of 0.88, outperforming baselines in detecting copyrighted books. We apply the optimal threshold of MIN-K% PROB to the test set of 10,000 snippets from 100 books from Books3. Table 2 represents the top 20 books with the highest predicted contamination rates. Figure 4 reveals nearly 90% of the books have an alarming contamination rate over 50%.

| Method | Book |
|---|---|
| Neighbor | 0.75 |
| PPL | 0.84 |
| Zlib | 0.81 |
| Lowercase | 0.80 |
| MIN-K% PROB | **0.88** |

Figure 3: AUC scores for detecting the validation set of copyrighted books on GPT-3.

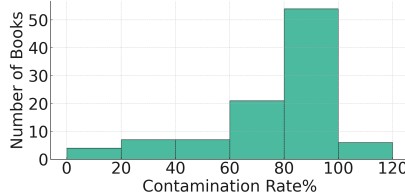

Figure 4: Distribution of detected contamination rate of 100 copyrighted books.

Table 2: Top 20 copyrighted books in GPT-3's pretraining data. The listed contamination rate represents the percentage of text excerpts from each book identified in the pretraining data.

| Contamination % | Book Title | Author | Year |
|---|---|---|---|
| 100 | The Violin of Auschwitz | Maria Àngels Anglada | 2010 |
| 100 | North American Stadiums | Grady Chambers | 2018 |
| 100 | White Chappell Scarlet Tracings | Iain Sinclair | 1987 |
| 100 | Lost and Found | Alan Dean | 2001 |
| 100 | A Different City | Tanith Lee | 2015 |
| 100 | Our Lady of the Forest | David Guterson | 2003 |
| 100 | The Expelled | Mois Benarroch | 2013 |
| 99 | Blood Cursed | Archer Alex | 2013 |
| 99 | Genesis Code: A Thriller of the Near Future | Jamie Metzl | 2014 |
| 99 | The Sleepwalker's Guide to Dancing | Mira Jacob | 2014 |
| 99 | The Harlan Ellison Hornbook | Harlan Ellison | 1990 |
| 99 | The Book of Freedom | Paul Selig | 2018 |
| 99 | Three Strong Women | Marie NDiaye | 2009 |
| 99 | The Leadership Mind Switch: Rethinking How We Lead in the New World of Work | D. A. Benton, Kylie Wright-Ford | 2017 |
| 99 | Gold | Chris Cleave | 2012 |
| 99 | The Tower | Simon Clark | 2005 |
| 98 | Amazon | Bruce Parry | 2009 |
| 98 | Ain't It Time We Said Goodbye: The Rolling Stones on the Road to Exile | Robert Greenfield | 2014 |
| 98 | Page One | David Folkenflik | 2011 |
| 98 | Road of Bones: The Siege of Kohima 1944 | Fergal Keane | 2010 |

# 6 CASE STUDY: DETECTING DOWNSTREAM DATASET CONTAMINATION

Assessing the leakage of downstream task data into pretraining corpora is an important issue, but it is challenging to address given the lack of access to pretraining datasets. In this section, we investigate the possibility of using MIN-K% PROB to detect information leakage and perform ablation studies to understand how various training factors impact detection difficulty. Specifically, we continually pretrain the 7B parameter LLaMA model (Touvron et al., 2023a) on pretraining data that have been purposefully contaminated with examples from the downstream task.

## 6.1 EXPERIMENTS

**Experimental setup.** To simulate downstream task contamination that could occur in real-world settings, we create contaminated pretraining data by inserting examples from downstream tasks into a pretraining corpus. Specifically, we sample text from the RedPajama corpus (TogetherCompute, 2023) and insert formatted examples from the downstream datasets BoolQ (Clark et al., 2019), IMDB (Maas et al., 2011), Truthful QA (Lin et al., 2021), and Commonsense QA (Talmor et al., 2019) in contiguous segments at random positions in the uncontaminated text. We insert 200 (positive) examples from each of these datasets into the pretraining data while also isolating a set of 200 (negative) examples from

each dataset that are known to be absent from the contaminated corpus. This creates a contaminated pretraining dataset containing 27 million tokens with 0.1% drawn from downstream datasets.

We evaluate the effectiveness of MIN-K% PROB at detecting leaked benchmark examples by computing AUC scores over these 400 examples on a LLaMA 7B model finetuned for one epoch on our contaminated pretraining data at a constant learning rate of 1e-4.

**Main results.** We present the main attack results in Table 3. We find that MIN-K% PROB outperforms all baselines. We report TPR@5%FPR in Table 6 in Appendix A, where MIN-K% PROB shows 12.2% improvement over the best baseline.

Table 3: AUC scores for detecting contaminant downstream examples. **Bold** shows the best AUC score within each column.

| Method | BoolQ | Commonsense QA | IMDB | Truthful QA | Avg. |
|--------|-------|----------------|------|-------------|------|
| Neighbor | 0.68 | 0.56 | 0.80 | 0.59 | 0.66 |
| Zlib | 0.76 | 0.63 | 0.71 | 0.63 | 0.68 |
| Lowercase | 0.74 | 0.61 | 0.79 | 0.56 | 0.68 |
| PPL | 0.89 | 0.78 | 0.97 | 0.71 | 0.84 |
| MIN-K% PROB | **0.91** | **0.80** | **0.98** | **0.74** | **0.86** |

## 6.2 RESULTS AND ANALYSIS

The simulation with contaminated datasets allows us to perform ablation studies to empirically analyze the effects of *dataset size*, *frequency of data occurrence*, and *learning rate* on detection difficulty, as theorized in section 2.1. The empirical results largely align with and validate the theoretical framework proposed. In summary, we find that detection becomes more challenging as data occurrence and learning rate decreases, and the effect of dataset size on detection difficulty depends on whether the contaminants are outliers relative to the distribution of the pretraining data.

**Pretraining dataset size.** We construct contaminated datasets of 0.17M, 0.27M, 2.6M and 26M tokens by mixing fixed downstream examples (200 examples per downstream task) with varying amounts of RedPajama data, mimicking real-world pretraining. Despite the theory suggesting greater difficulty with more pretraining data, Figure 5a shows AUC scores counterintuitively increase with pre-training dataset size. This aligns with findings that LMs better memorize tail outliers (Feldman, 2020; Zhang et al., 2021). With more RedPajama tokens in the constructed dataset, downstream examples become more significant outliers. We hypothesize that their enhanced memorization likely enables easier detection with perplexity-based metrics.

To verify the our hypothesis, we construct control data where contaminants are not outliers. We sample Real Time Data News August 2023[4], containing post-2023 news absent from LLaMA pretraining. We create three synthetic corpora by concatenating 1000, 5000 and 10000 examples from this corpus, hence creating corpora of sizes 0.77M, 3.9M and 7.6M tokens respecitvely. In each setting, we consider 100 of these examples to be contaminant (positive) examples and set aside another set of 100 examples from News August 2023 (negative). Figure 5b shows AUC scores decrease as the dataset size increases.

Detection of outlier contaminants like downstream examples gets easier as data size increases, since models effectively memorize long-tail samples. However, detecting general in-distribution samples from the pretraining data distribution gets harder with more data, following theoretical expectations.

**Data occurrence.** To study the relationship between detection difficulty and data occurrence, we construct a contaminated pretraining corpus by inserting multiple copies of each downstream data point into a pre-training corpus, where the occurrence of each example follows a Poisson distribution. We measure the relationship between the frequency of the example in the pretraining data and its AUC scores. Figure 5c shows that AUC scores positively correlates with the occurrence of examples.

---

[4]`https://huggingface.co/datasets/RealTimeData/News_August_2023`

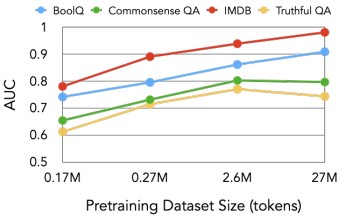 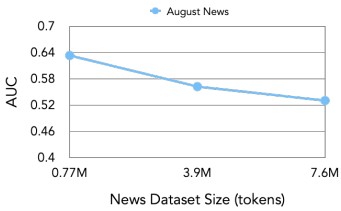 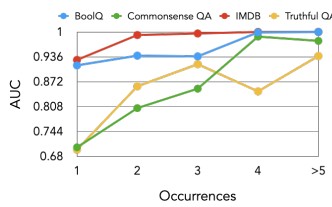

(a) Outlier contaminants, e.g., downstream examples, become easier to detect as dataset size increases.

(b) In-distribution contaminants, e.g., news articles, are harder to detect as dataset size increases.

(c) Contaminants that occur more frequently in the dataset are easier to detect.

Figure 5: We show the effect of contamination rate (expressed as a percentage of the total number of pretraining tokens) and occurrence frequency on the ease of detection of data contaminants using MIN-K% PROB.

**Learning rate.** We also study the effect of varying the learning rates used during pretraining on the detection statistics of the contaminant examples (see Table 4). We find that raising the learning rate from $10^{-5}$ to $10^{-4}$ increases AUC scores significantly in all the downstream tasks, implying that higher learning rates cause models to memorize their pretraining data more strongly. A more in-depth analysis in Table 7 in Appendix A demonstrates that a higher learning rate leads to more memorization rather than generalization for these downstream tasks.

Table 4: AUC scores for detecting contaminant downstream examples using two different learning rates. Detection becomes easier when higher learning rates are used during training. **Bold** shows the best AUC score within each column.

| Learning rate | BoolQ | Commonsense QA | IMDB | LSAT QA | Truthful QA |
|---------------|-------|----------------|------|---------|-------------|
| $1 \times 10^{-5}$ | 0.64 | 0.59 | 0.76 | 0.72 | 0.56 |
| $1 \times 10^{-4}$ | **0.91** | **0.80** | **0.98** | **0.82** | **0.74** |

# 7 RELATED WORK

**Membership inference attack in NLP.** Membership Inference Attacks (MIAs) aim to determine whether an arbitrary sample is part of a given model's training data (Shokri et al., 2017; Yeom et al., 2018b). These attacks pose substantial privacy risks to individuals and often serve as a basis for more severe attacks, such as data reconstruction (Carlini et al., 2021; Gupta et al., 2022; Cummings et al., 2023). Due to its fundamental association with privacy risk, MIA has more recently found applications in quantifying privacy vulnerabilities within machine learning models and in verifying the accurate implementation of privacy-preserving mechanisms (Jayaraman & Evans, 2019; Jagielski et al., 2020; Zanella-Béguelin et al., 2020; Nasr et al., 2021; Huang et al., 2022; Nasr et al., 2023; Steinke et al., 2023). Initially applied to tabular and computer vision data, the concept of MIA has recently expanded into the realm of language-oriented tasks. However, this expansion has predominantly centered around finetuning data detection (Song & Shmatikov, 2019; Shejwalkar et al., 2021; Mahloujifar et al., 2021; Jagannatha et al., 2021; Mireshghallah et al., 2022b). Our work focuses on the application of MIA to pretraining data detection, an area that has received limited attention in previous research efforts.

# 8 CONCLUSION

We present a pre-training data detection dataset WIKIMIA and a new approach MIN-K% PROB. Our approach uses the intuition that trained data tends to contain fewer outlier tokens with very low probabilities compared to other baselines. Additionally, we verify the effectiveness of our approach in real-world setting, we perform two case studiies: detecting dataset contamination and published book detection. For dataset contamination, we observe empirical results aligning with theoretical predictions about how detection difficulty changes with dataset size, example frequency, and learning rate. Most strikingly, our book detection experiments provide strong evidence that GPT-3 models may have been trained on copyrighted books.

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

# A    ADDITIONAL RESULTS

Table 5: TPR@5%FPR score for detecting pretraining examples from the given model on WIKIMIA for MIN-K% PROB and baselines. *Ori.* and *Para.* denote the original and paraphrase settings, respectively. **Bold** shows the best score within each column.

| Method | Pythia-2.8B | | NeoX-20B | | LLaMA-30B | | LLaMA-65B | | OPT-66B | | |
| | *Ori.* | *Para.* | *Ori.* | *Para.* | *Ori.* | *Para.* | *Ori.* | *Para.* | *Ori.* | *Para.* | Avg. |
|---|---|---|---|---|---|---|---|---|---|---|---|
| Neighbor | 10.2 | 16.2 | 15.2 | 19.3 | 20.1 | 17.2 | 17.2 | 20.0 | 17.3 | 18.8 | 17.2 |
| PPL | 9.4 | 18.0 | 17.3 | 24.9 | 23.7 | 18.7 | 16.5 | 23.0 | 20.9 | 20.1 | 19.3 |
| Zlib | 18.7 | 18.7 | 20.3 | 22.1 | 18.0 | 20.9 | 23.0 | 23.0 | 21.6 | 20.1 | 20.6 |
| Lowercase | 10.8 | 7.2 | 12.9 | 12.2 | 10.1 | 6.5 | 14.4 | 12.2 | 14.4 | 8.6 | 10.9 |
| Smaller Ref | 10.1 | 10.1 | 15.8 | 10.1 | 10.8 | 11.5 | 15.8 | 21.6 | 15.8 | 10.1 | 13.2 |
| MIN-K% PROB | 13.7 | 15.1 | 21.6 | 27.3 | 22.3 | 25.9 | 20.9 | 30.9 | 21.6 | 23.0 | 22.2 |

Table 6: TPR @ FPR=5% for detecting contaminant downstream examples using reference-based and reference-free methods. **Bold** shows the best reference-free TPR within each column.

| Method | BoolQ | Commonsense QA | IMDB | Truthful QA | Avg. |
|---|---|---|---|---|---|
| Neighbor | 19 | 7 | 41 | 13 | 20 |
| PPL | 52 | 24 | 74 | 17 | 42 |
| Zlib | 18 | 9 | 19 | 7 | 13 |
| Lowercase | 24 | 3 | 26 | 14 | 17 |
| MIN-K% PROB | 55 | 23 | 83 | 21 | 46 |

Table 7: Accuracy of the model finetuned in Section 6.1 on each non-contaminant and contaminant examples used for AUC computation for each downstream dataset. The difference in average classification accuracy of contaminant examples over that of non-contaminant examples is 0.04 at a learning rate of $1 \times 10^{-5}$ and 0.11 at a learning rate of $1 \times 10^{-4}$. This indicates that memorization becomes a significantly more pronounced effect than generalization at larger learning rates.

| Learning rate | BoolQ | Commonsense QA | IMDB | LSAT QA | Truthful QA | Avg. |
|---|---|---|---|---|---|---|
| *Non-contaminant examples* | | | | | | |
| $1 \times 10^{-5}$ | 0.68 | 0.47 | 0.89 | 0.22 | 0.28 | 0.51 |
| $1 \times 10^{-4}$ | 0.69 | 0.48 | 0.90 | 0.24 | 0.33 | 0.53 |
| *Contaminant examples* | | | | | | |
| $1 \times 10^{-5}$ | 0.71 | 0.49 | 0.92 | 0.26 | 0.38 | 0.55 |
| $1 \times 10^{-4}$ | 0.81 | 0.60 | 0.89 | 0.35 | 0.56 | 0.64 |

Table 8: Input template we use to prompt GPT-4 to obtain the similarity score.

| System | You are a helpful assistant in evaluating the similarity between two outputs generated by two different AI chatbots. Your goal is to rate the similarity between the two outputs based on a scale of 1 to 5, with 1 being highly dissimilar and 5 being highly similar. |
|---|---|
| User | Rate the similarity between Output (a) and Output (b) on a scale of 1 to 5, where 1 indicates high dissimilarity, and 5 indicates high similarity. Here are some rules of the evaluation: (1) Consider how closely Output (a) matches Output (b) in terms of content, context, and relevance. (2) Do not provide a rating outside the 1 to 5 scale, and avoid giving a rating of 3 (neutral) whenever possible. (3) Your judgment should be as objective as possible, without being influenced by any potential bias. You should answer 'Score: ', followed by an integer rating between 1 to 5, where 1 indicates high dissimilarity, and 5 indicates high similarity. You should then output 'Reason: ' and provide a short sentence to explain your rating. Output (a): Output (b): |

Table 9: Examples of similar ground truth completion and the completion by the `LLaMA2-7B-WhoIsHarryPotter` model, alongside the SimCSE and GPT-4 similarity scores.

**Chunk #1:**
▶ Context: He was chewing the gillyweed as hard and fast as he could; it felt unpleasantly slimy and rubbery, like octopus tentacles. Waist-deep in the freezing water he stopped, swallowed, and waited for something to happen. He could hear laughter in the crowd and knew he must look stupid, walking into the lake without showing any sign of magical power. The part of him that was still dry was covered in goose pimples; half immersed in the icy water, a cruel breeze lifting his hair, Harry started to shiver violently. He avoided looking at the stands; the laughter was becoming louder, and there were catcalls and jeering from the Slytherins. ... Then, quite suddenly, Harry felt as though an invisible pillow had been pressed over his mouth and nose. He tried to draw breath, but it made his head spin; his lungs were empty, and he suddenly felt a piercing pain on either side of his neck - Harry clapped his hands around his throat and felt two large slits just below his ears, flapping in the cold air. . . . He had gills. Without pausing to think, he did the only thing that made

▶ Ground truth (Output a): sense - he flung himself forward into the water. The first gulp of icy lake water felt like the breath of life. His head had stopped spinning; he took another great gulp of water and felt it pass smoothly through his gills, sending oxygen back to his brain. He stretched out his hands in front of him and stared at them. They looked green and ghostly under the water, and they had become webbed. He twisted around and looked at his bare feet - they had become elongated and the toes were webbed too: It looked as though he had sprouted flippers. The water didn't feel icy anymore either ... on the contrary, he felt pleasantly cool and very light. . . . Harry struck out once more, marveling at how far and fast his flipper-like feet propelled him through the vater, and noticing how clearly he could see, and how he no longer seemed to need to blink. He had soon swum so far into the lake that he could no longer see the bottom. He flipped over and dived into its depths. Silence pressed upon his ears as he soared over a strange, dark, foggy landscape. He could only see ten feet around him, so that as he sped throuugh the water new scenes seemed to loom suddenly out of the incoming darkness: forests of rippling, tangled black weed, wide plains of mud littered with dull, glimmering stones. He swam deeper and deeper, out toward the middle of the lake, his eyes wide, staring through the eerily gray-lit water around him to the shadow beyond, where the water became opaque. Small fish flickered past him like silver darts. Once or twice he thought he saw something larger moving ahead of him, but when he got nearer, he discovered it to be

▶ Completion by `LLaMA2-7B-WhoIsHarryPotter` (Output b): sense: he opened his mouth and let the water rush in.
Reflexively, he gulped air, and his lungs expanded, filling with the icy water. The coldness spread through his body, and he shivered again, but this time it was different. He felt a strange tingling sensation, like tiny fingers massaging his muscles.
As he gulped in more water, his body began to change. His legs grew longer and thinner, turning into a pair of flippers. His arms shortened and transformed into a pair of fins, and his hands became webbed.

▶ SimCSE score: 0.82

▶ GPT-4 score and reason: 5. Both outputs describe a character changing into a sea creature. They both mention the character gulping in water, feeling cold initially then adjusting to the temperature, and developing webbed hands and feet or flippers. However, Output (b) doesn't include the exploration of the underwater environment that is a key part of Output (a) which results in slight difference.

**Chunk #2:**

► Context: doesn't know about the troll." Ron bit his lip. "Oh, all right," he snapped. "But Percy'd better not see us." Ducking down, they joined the Hufflepuffs going the other way, slipped down a deserted side corridor, and hurried off toward the girls' bathroom. They had just turned the corner when they heard quick footsteps behind them. "Percy!" hissed Ron, pulling Harry behind a large stone griffin. Peering around it, however, they saw not Percy but Snape. He crossed the corridor and disappeared from view. "What's he doing?" Harry whispered. "Why isn't he down in the dungeons with the rest of the teachers?" "Search me." Quietly as possible, they crept along the next corridor after Snape's fading footsteps. "He's heading for the third floor," Harry said, but Ron held up his hand. "Can you smell something?" Harry sniffed and a foul stench reached his nostrils, a mixture of old socks and the kind of public toilet no one seems to clean. And then they heard it – a low grunting, and the shuffling footfalls of gigantic feet. Ron pointed – at the end of a passage to the left, something huge was moving toward them. They shrank into the shadows and

► Ground truth: watched as it emerged into a patch of moonlight. It was a horrible sight. Twelve feet tall, its skin was a dull, granite gray, its great lumpy body like a boulder with its small bald head perched on top like a coconut. It had short legs thick as tree trunks with flat, horny feet. The smell coming from it was incredible. It was holding a huge wooden club, which dragged along the floor because its arms were so long. The troll stopped next to a doorway and peered inside. It waggled its long ears, making up its tiny mind, then slouched slowly into the room. "The keys in the lock," Harry muttered. "We could lock it in." "Good idea," said Ron nervously. They edged toward the open door, mouths dry, praying the troll wasn't about to come out of it. With one great leap, Harry managed to grab the key, slam the door, and lock it. 'Yes!" Flushed with their victory, they started to run back up the passage, but as they reached the corner they heard something that made their hearts stop – a high, petrified scream – and it was coming from the chamber they'd just chained up. "Oh, no," said Ron, pale as the Bloody Baron. "It's the girls' bathroom!" Harry gasped. "Hermione!" they said together. It was the last thing they wanted to do, but what choice did they have? Wheeling around, they sprinted back to the door and turned the key, fumbling in their panic. Harry pulled the door open and they ran inside. Hermione Granger was shrinking against the wall opposite, looking as if she was about to faint. The troll was advancing on her, knocking the sinks off the walls as it went. "Confuse it!" Harry said desperately to Ron, and, seizing a tap, he threw it as hard as he could against the wall.

► Completion by LLaMA2-7B-WhoIsHarryPotter (Output b): watched in horror as the creature came into view. It was a troll from the depths of the earth, its skin the color of dirt and mold, its eyes glowing red slits in the dim light. The passageway was too narrow for it to fit, but it could see them, and its grunts grew louder, more menacing. "What do we do?" gulped Harry. "We can't outrun it." "We have to find a way to distract it," decided Ron, his mind racing with ideas. "Quick!" He grabbed a nearby book from the wall and flung it at the troll, hoping to divert its attention. The creature roared in anger and chased after the book, giving them just enough time to slip through a nearby door and slam it shut behind them. They leaned against the door, panting with fear, and listened as the troll beat its fists against the other side of the door, trying to break through. "That was close," said Harry, his voice shaking. "Too close." "We have to get out of here," agreed Ron. "Before it breaks through." But they knew they couldn't escape that way. The only other exit was through the library, and they knew Mr. Hufflepuff would never let them pass. They were trapped.

► SimCSE score: 0.84

► GPT-4 score and reason: 4. Both outputs present a similar scenario where characters are encountered by a troll and try to escape or distract it. The issue they face, the settings, and the way they express fear show considerable similarities. However, the specific details and descriptions of events do somewhat differ.

**Chunk #3:**

► Context: one, 'cause it might be the last thing yeh do." Malfoy, Crabbe, and Goyle weren't listening; they were talking in an undertone and Harry had a nasty feeling they were plotting how best to disrupt the lesson. "Yeh always wait fer the hippogriff ter make the firs' move," Hagrid continued. "It's polite, see? Yeh walk toward him, and yeh bow, an' yeh wait. If he bows back, yeh're allowed ter touch him. If he doesn' bow, then get away from him sharpish, 'cause those talons hurt. "Right – who wants ter go first?" Most of the class backed farther away in answer. Even Harry, Ron, and Hermione had misgivings. The hippogriffs were tossing their fierce heads and flexing their powerful wings; they didn't seem to like being tethered like this. "No one?" said Hagrid, with a pleading look. "I'll do it," said Harry. There was an intake of breath from behind him, and both Lavender and Parvati whispered, "Oooh, no, Harry, remember your tea leaves!" Harry ignored them. He climbed over the paddock fence. "Good man, Harry!" roared Hagrid. "Right then – let's see how yeh get on with Buckbeak." He untied one of the chains, pulled the gray hippogriff

► Ground truth (Output a): "away from its fellows, and slipped off its leather collar. The class on the other side of the paddock seemed to be holding its breath. Malfoy's eyes were narrowed maliciously. "Easy) now, Harry," said Hagrid quietly. "Yeh've got eye contact, now try not ter blink.... Hippogriffs don' trust yeh if yeh blink too much...." Harry's eyes immediately began to water, but he didn't shut thern. Buckbeak had turned his great, sharp head and was staring at Harry with one fierce orange eye. "Tha's it," said Hagrid. "Tha's it, Harry... now, bow." Harry didn't feel much like exposing the back of his neck to Buckbeak, but he did as he was told. He gave a short bow and then looked up. The hippogriff was still staring haughtily at him. It didn't move. "Ah," said Hagrid, sounding worried. "Right – back away, now, Harry, easy does it But then, to Harry's enormous surprise, the hippogriff suddenly bent its scaly front knees and sank into what was an unmistakable bow. "Well done, Harry!" said Hagrid, ecstatic. "Right – yeh can touch him! Pat his beak, go on!" Feeling that a better reward would have been to back away, Harry moved slowly toward the hippogriff and reached out toward it. He patted the beak several times and the hippogriff closed its eyes lazily, as though enjoying it. The class broke into applause, all except for Malfoy, Crabbe, and Goyle, who were looking deeply disappointed. "Righ' then, Harry," said Hagrid. "I reckon he might' let yeh ride him!" This was more than Harry had bargained for. He was used to a broomstick; but he wasn't sure a hippogriff would be quite the same. "Yeh climb up there, jus' behind the wing joint," said Hagrid, "an' mind yeh don' pull any of his feathers out, he won' like that...." Harry put his foot on the top of Buckbeaks

► Completion by LLaMA2-7B-WhoIsHarryPotter (Output b): forward, and held it out on a leash. The bird was massive, with a beak like a scythe and eyes that glowed like lanterns in the dark. "Walk towards him, boys and girls," urged Hagrid. "Be polite, now. No need ter be afraid. He'n't goin ter hurt ye." But he wasn't sure if that was true. He could feel a dark mischief in the air, a sense of unease that made his spine tingle. He'd heard stories of hippogriffs turning on their handlers when they were least expected, and he didn' mind sharing his tea leaves with them. But he couldn' afford ter back down now. So he took a deep breath and started walking, trying ter keep his eyes fixed on the bird' eyes. "Good boy, good boy," coaxed Haggridd. "Now, touch his wing, if yeh dare."

► SimCSE score: 0.77

► GPT-4 score and reason: 4. The two outputs share notable similarities in context, content, and elements. They both discuss Hagrid instructing someone to approach a hippogriff, with an emphasis on maintaining eye contact and eventual touching. While differences can be seen in the detailed dialogues or descriptions, the general themes and ideas remain consistent.

## B  DETAILS OF WIKIMIA

**Data properties.**   Our WIKIMIA benchmark demonstrates several desirable properties that make it suitable for evaluating methods to detect data during pretraining on any newly released models.

(1) *Accurate:* Since non-member data consists of events that occurred after the LM pretraining, there is a guarantee that this data was not present during pretraining, ensuring the accuracy of our dataset. We consider Wikipedia event data because of its time sensitivity. A recent non-event Wikipedia page may be only a recent version of an older page that was already present during the model's pretraining, and thus it may not serve as true non-member data. For example, a Wikipedia page created after 2023 about a historical figure or a well-known concept could contain substantial text already mentioned in the pretraining corpus.

(2) *General:* Our benchmark is designed to be widely applicable across different models pretrained on Wikipedia, a commonly used source of pretraining data. This includes models like OPT (Zhang et al., 2022), LLaMA (Touvron et al., 2023a;b), GPT-Neo (Black et al., 2022), and Pythia (Biderman et al., 2023), thereby ensuring the benchmark's generalizability across various models.

(3) *Dynamic:* Our benchmark will be continually updated by incorporating the latest non-member data, such as recent events from Wikipedia. This consistent renewal ensures that the benchmark's

non-member data is always up-to-date and can be used to evaluate MIA for any newly introduced pretrained models.

## C  DETAILS OF MIN-K% PROB

---
**Algorithm 1** Pretraining Data Detection
---
1: **Input:** A sequence of tokens $x = x_1, x_2, ..., x_N$, decision threshold $\epsilon$
2: **Output:** Membership of the sequence $x$
3: **for** $i = 1$ to $N$ **do**
4:     Compute $-\log p(x_i | x_1, \ldots, x_{i-1})$
5: **end for**
6: Select the top $k\%$ of tokens from $x$ with the lowest probability and add to Min-k%$(x)$
7: MIN-K% PROB$(x) = \sum_{x_i \in \text{Min-k\%}(x)} -\log p(x_i | x_1, ..., x_{i-1})$
8: **If** MIN-K% PROB$(x) > \epsilon$ **: return** Non-member      **Else: return** Member
---

