# OpenReview forum: "Detecting Pretraining Data from Large Language Models"
_ICLR.cc/2024/Conference — ICLR 2024 poster_

### Official Review · Reviewer_oVRd · 2023-10-18

**Soundness:** 3 good
**Presentation:** 3 good
**Contribution:** 2 fair
**Rating:** 5
**Confidence:** 3

**Summary:**

The paper presents a method and benchmark to detect if a type of data appear in LLM pretraining. The main idea is to use the average of low probably of n (20 in the paper) tokens to gauge if the piece of text is used in pretraining. Paper is well-written and the idea is easy to follow.

**Strengths:**

- The author touched an interesting topic. The experiment looks solid. Paper is well-written and easy to follow.

**Weaknesses:**

- The methodology sounds dubious. It doesn't take consideration of the data ordering in the training phase. For instance, in the paper, authors showed that learning rate affects the result a lot. In the late stage of training, the learning rate will be small which could impact how well model remember the data.
- It also doesn't consider the data distribution/mixture in the training. e.g. llama training duplicates wikipedia data.
- High probability of a small piece of text doesn't mean that model is trained entirely on the content. The piece of text might appear somewhere else.
- Lack enough of novelty

**Questions:**

- In section 5.2, it should be 50% of books have over 90% contamination?
- The method used here doesn't take consideration of the data ordering in the training phase. For instance, as is shown in the paper that learning rate affects the prediction a lot. In the late stage of training, the learning rate would be small which could impact how well model remember the data.

---

> ### Author Response · Authors · 2023-11-21
> **Response to Reviewer oVRd**
>
> We thank the reviewer for their constructive comments/feedback. We respond to the reviewer’s comments and questions below.
>
> ---
>
> > The methodology sounds dubious. It doesn't take consideration of the data ordering in the training phase. For instance, in the paper, authors showed that learning rate affects the result a lot. In the late stage of training, the learning rate will be small which could impact how well model remember the data. It also doesn't consider the data distribution/mixture in the training. e.g. llama training duplicates wikipedia data.
>
> We find the reviewer has some misconceptions of our method and setting. As clearly stated in abstract/intro and other sections, we study the pretraining data detection problem of **black-box language models: we only have black-box access to an LLM without knowing the pretraining data. It means we don’t know the data ordering, learning rates, and pretraining data distribution of the target model.** We study the problem under this black-box assumption because state-of-art LMs such as OpenAI GPT does not reveal any detail of their pretraining process and pretraining data. We show our method could be effectively applied to detecting copyrighted books in OpenAI text-davinci-003 (in Section 6).
>
> Our main results in Table 1 are a fair comparison with baseline approaches using the *same model*. The learning rates and data factors are the same for each of these comparisons. Consequently, the fact that Min-K Prob provides a consistent margin regardless of the source model shows that it robustly works better than other approaches. Note that we are not claiming Min-K prob can solve the training detection problem perfectly in any scenario (such as the low learning rate scenario as the author pointed out), but Min-K prob is a better detection method than existing methods.
>
> ---
>
> > High probability of a small piece of text doesn't mean that model is trained entirely on the content. The piece of text might appear somewhere else.
>
> We would like to clarify some misunderstandings regarding our approach. Our method is based on the hypothesis that an unseen example is likely to contain a few outlier words with low probabilities under the LLM, while a seen example is less likely to have words with such low probabilities. This is a key distinction – **our focus is on the identification of texts with extremely low probability words, rather than high probability ones.** In other words, our method identifies a text segment as a potential seen example if a significant portion of text exhibits non-low probabilities, rather than *"a small piece of text has high probability"*. Furthermore,  we have systematically validated it across seven different datasets with known ground truths. These datasets have been meticulously constructed to include both seen and unseen data, verified to be either included or excluded from the model’s pretraining corpus. Our results demonstrate that our method consistently outperforms previous baselines.
>
> > Lack enough of novelty
>
>
> We respectfully request that the reviewer provide further details regarding their concerns about the lack of novelty in our work. Could you direct us to any related works for comparison?
>
> We are **the first to study the pretraining data detection problem of large-scale black-box language models.** We highlight three primary areas of innovation:
>
> * Creation of a Dynamic Benchmark: We introduce a dynamic benchmark specifically designed for this new problem domain. This benchmark is the first of its kind and is tailored to assess the effectiveness of data detection methods in the context of large-scale language models.
>
> * Development of the detection method MIN-K% PROB: Our proposed method, MIN-K% PROB, is a significant departure from existing detection techniques. Unlike previous methods that require training a reference model on data akin to the pretraining corpus, MIN-K% PROB requires no knowledge of the pretraining data and no additional training. We demonstrate the superior performance of our methods compared to existing baselines across all datasets.
>
> * Real-World Application: We apply our research to real-world applications, including the detection of copyrighted books and contaminated downstream examples. These case studies showcase their practical relevance in addressing critical issues in the field.
>
> ---
>
> > In section 5.2, it should be 50% of books have over 90% contamination?
>
> In section 5.2, we claim nearly 90% of the books have an alarming contamination rate over 50%, which is accurate.

---

> ### Author Response · Authors · 2023-11-22
> **Follow up on our previous response**
>
> Thank you again for your review and feedback! As the discussion period is coming to an end, we want to check in and see if our previous response has addressed your concerns. If you have any follow-up questions or any concerns we haven't addressed yet for a better score, please let us know and we would be happy to answer them.

---

> > ### Comment · Reviewer_oVRd · 2023-12-04
> > **Reply to author response**
> >
> > The clarification here doesn't address my concern that when the test piece of text showed low probability, can we guarantee that it's the outlier and haven't been trained on? The paper seems only taking consideration of memorization but not forgetting process and generalization in model training. The training dynamics matter here (such as learning rate, data ordering etc). Remember that today's LLMs could be a multi-stage training model (e.g. CodeLLaMA). Is there a way to reduce false negatives? That's what I want to see. It could be a much harder problem though.

---

### Official Review · Reviewer_syso · 2023-10-29

**Soundness:** 3 good
**Presentation:** 3 good
**Contribution:** 2 fair
**Rating:** 6
**Confidence:** 4

**Summary:**

The paper is about speculating whether a piece of text has appeared in the pretraining corpus of an LLM or not. Authors prepare a corpus by crawling wikipedia. Their idea is to partition event pages into those before a certain date and those after the date, to obtain pages that have been or not been used in pretraining corpora.

They also report that for the texts that have not been used during the pretraining stage, word occurrence probabilities are lower, which is a simple heuristic.

Most of the paper is dedicated to reporting various creative experiments. Apart from experiments on the corpus, and on the proposed heuristic; a set of experiments on speculating the use of copyrighted books in the pretraining corpora of existing LLMs, and also another set of experiments on the possibility of detecting leaked training sets into the pretraining corpus.

**Strengths:**

Except in a handful of cases, the paper is well-written.\
A dataset will be released.\
The proposed heuristic works.

**Weaknesses:**

Not trying to undermine the amount of work done by the authors, but I feel the paper is packed with distracting experiments. I think the two case studies reported by the authors, which have taken up three pages, should be two blog posts, rather than three pages of an ICLR paper. In my opinion it is fine that the proposed method (or heuristic) is simple. But it is not fine to fill out the remaining pages with experiments which are either judgmental (the first case study) or repetitious (the second case study).

Page 8, second paragraph. Authors report that their accuracy is proportional to the pretraining corpus size. They also cite a paper on memorization of NNs, and argue that outliers are memorized better.\
My question is: with the same theory, can Authors justify why the outliers of a small dataset are not memorized better than the outliers of a large dataset? Because that is what they are reporting. This cannot be supported by the cited theory.

My last question (and the most important one) is that why the proposed method works? Authors have not answered this pivotal question. Why not using the occurrence probability of all the words, why not the occurrence probability of the entire sentence, why not the occurrence probability of the sentences that appear in the pretraining corpus and filtering out the rest. "WHY" the proposed heuristic works?


= = = = = = = = = = = = = = = \
The comment below was added after I reviewed the paper and posted the score, but before it was released to the authors. So it had no effect on my review and the review score.

I suggest the authors look into the paper below---published about 10 days ago---the paper claims that LLMs memorize training sets, although not verbatim. So the proposed method by the authors will fail to detect such cases---I know that the authors have said they would only look into the exact match case.\
[*] Elephants Never Forget: Testing Language Models for Memorization of Tabular Data. Sebastian Bordt, Harsha Nori, Rich Caruana. NeurIPS 2023 Second Table Representation Learning Workshop.

**Questions:**

Page 3, Paragraph 6, what is x? It looks to be out of context.

In Section 6.1, you have a passage that starts with “main results”, and then after that you have another section called “Results and …”. Please merge these two.

Section 3, why are you using “negative log-likelihood”? In optimization people use it, because the derivative operator minimizes the loss function. You are not minimizing anything, so you do not need to say “negative …” to be forced to also say “the highest negative ….”. The same thing applies to Page 15, in your pseudo-code. Just say log-likelihood.

In Page 8, paragraph “Data occurrence”, it looks Authors are not aware that this is not a new finding. [1] reports comprehensive experiments on this subject.


[1] Large Language Models Struggle to Learn Long-Tail Knowledge. ICML 2023.

---

> ### Author Response · Authors · 2023-11-21
> **Response to Reviewer syso**
>
> We thank the reviewer for their constructive comments/feedback. We respond to the reviewer’s comments and questions below.
>
> ---
>
> >  It is not fine to fill out the remaining pages with experiments which are either judgmental (the first case study) or repetitious (the second case study).
>
> We strongly disagree that our first case study is judgemental and the second case study is repetitious.
>
> Regarding the first case study, it is designed to demonstrate the effectiveness of our methods in detecting pretraining data from real-world black-box language models, such as OpenAI's GPT-3.5. This directly aligns with our primary research objective. To ensure objectivity, we carefully constructed a validation dataset comprising copyrighted books proved to be included in GPT's pretraining data as member data and new books data as non-member. This approach allows us to rigorously validate our method's effectiveness. We have also released this dataset for public use, enabling the evaluation of detection methods for pretraining data.
>
> As for the second case study, it offers novel insights into factors influencing pretraining data detection. It shows how detection difficulty increases with increased training data sizes and reduced occurrence frequencies of detecting examples, along with the impact of varying learning rates. These insights are crucial for understanding model memorization behaviors.
>
> ---
>
> > Page 8, second paragraph. Authors report that their accuracy is proportional to the pretraining corpus size. They also cite a paper on memorization of NNs, and argue that outliers are memorized better. My question is: with the same theory, can Authors justify why the outliers of a small dataset are not memorized better than the outliers of a large dataset? Because that is what they are reporting. This cannot be supported by the cited theory.
>
> Thanks for the insightful question! To clarify, Feldman et al., 2021 [1] state that the learning algorithms tend to memorize the outliers for a better generalization. Zhang et al. 2021 [2] use counterfactual memorization to capture the source of memorization at test time and empirically identify a long tail of examples (i.e., outliers) with high memorization scores.
>
> To connect them to our experiment, our hypothetical explanation is as follows. In a larger dataset, downstream data, comprising a smaller fraction (0.01%), more resembles an outlier distribution and, hence, might be memorized more effectively compared to its proportion in a smaller dataset (10% downstream data). Although Feldman et al., 2021[1] did not explicitly mention this observation, our additional experiments with in-distribution data support our hypothesis, showing a reversal of dataset effect. **We have updated the paper (Section 6.2)** to state more clearly about the connection to Feldman et al., 2021[1] (we've highlighted the changes in red)
>
> [1] Vitaly Feldman. Does learning require memorization? a short tale about a long tail. In Proceedings of the 52nd Annual ACM SIGACT Symposium on Theory of Computing, pp. 954–959, 2020.
>
> [2] Chiyuan Zhang, Daphne Ippolito, Katherine Lee, Matthew Jagielski, Florian Tramèr, and Nicholas Carlini. Counterfactual memorization in neural language models. arXiv preprint arXiv:2112.12938, 2021
>
> ---
>
> > I suggest the authors look into the paper below---published about 10 days ago---the paper claims that LLMs memorize training sets, although not verbatim. So the proposed method by the authors will fail to detect such cases---I know that the authors have said they would only look into the exact match case.
>
> We would like to clarify that our study **is not limited to detecting exact match cases**. As explicitly stated in Section 2.2 under 'benchmark setting', our research addresses the detection of pretraining data in both exact match and **paraphrased scenarios**. As shown in Table 1, our method is effective for detecting paraphrases of training examples. This demonstrates our approach's capability to identify variations of training data, not just verbatim instances. Thanks for pointing us to the related works. **We have included discussion of this paper in our updated version** (Section 2.2, highlighted in yellow).

---

> ### Author Response · Authors · 2023-11-21
> **Response to Reviewer syso: following the previous response**
>
> > My last question (and the most important one) is that why the proposed method works? Authors have not answered this pivotal question. Why not using the occurrence probability of all the words...
>
> Our method is based on the intuition that trained data tends to contain fewer outlier tokens with very low probabilities compared to other baselines. If a sequence has been trained, gradient descent on this sequence almost ensures a reduction in the losses of all tokens, making it unlikely for tokens with extremely low probabilities to exist under the trained language model. Conversely, untrained sequences are prone to contain unexpected tokens with lower probabilities, although these sequences are still valid.
>
> ---
>
> > Page 3, Paragraph 6, what is x? It looks to be out of context.
>
> As defined in the first paragraph on Page 3, we consider 'x' to be a data point for which we seek to determine whether it is part of the pretraining data. **We have made a change to the paper to make it more clear (Section 2.1).**
>
> ---
>
> > In Page 8, paragraph “Data occurrence”, it looks Authors are not aware that this is not a new finding. [1] reports comprehensive experiments on this subject.
>
> Thank you for sharing [1]. **We have incorporated it into our submission when discussing our findings (Section 6.2, highlighted in yellow)** It's worth noting that we arrived at this conclusion through a different approach compared to [1]: While [1] measures memorization by conducting a data extraction attack [2] and reporting the reconstruction fidelity, we utilize a membership inference attack. This divergence in methodologies, yet convergence in findings, also adds to the credibility of our results.
>
> Furthermore, our study extends beyond data occurrences. We conduct additional experiments analyzing how various factors, such as learning rates and dataset size, influence the overall performance. This broader scope allows us to offer a more comprehensive perspective on how the pretraining design choices affects detection difficulty.
>
> [1] Large Language Models Struggle to Learn Long-Tail Knowledge. ICML 2023.
> [2] Carlini, et al. "Extracting training data from large language models." USENIX Security 2021.
>
> ---
>
> > In Section 6.1, you have a passage that starts with “main results”, and then after that you have another section called “Results and …”. Please merge these two.
>
> Thanks for pointing it out. We have incorporated a change into the paper.

---

> ### Author Response · Authors · 2023-11-22
> **Follow-up on our previous response**
>
> Thank you again for your review and feedback! As the discussion period is coming to an end, we want to check in and see if our previous response has addressed your concerns. If you have any follow-up questions or any concerns we haven't addressed yet for a better score, please let us know and we would be happy to answer them.

---

> ### Comment · Reviewer_syso · 2023-12-01
>
> thanks for the response. i don't agree with you that the cases studies are helpful, all the experiments could have been reported using the main datasets used to report the main results. this coupled with the fact that you have not explored exactly why your solution works and the alternatives may not work, and instead resorted to an intuition, makes the paper content very distracting. the current revision more looks like an entertaining magazine, rather than a paper that answers real questions. After all, what does the sentence below mean, which you used in your response:
> > ... detecting pretraining data from real-world black-box language models ...
>
> Have you done any experiments to support the claim that there is a difference between "real-world" and "academic" language models?
>
> **
>
> i am updating my score from marginally below to marginally above

---

### Official Review · Reviewer_S4cH · 2023-10-31

**Soundness:** 3 good
**Presentation:** 4 excellent
**Contribution:** 4 excellent
**Rating:** 8
**Confidence:** 3

**Summary:**

In this work, the authors primarly focus on tackling the problem of detecting examples included in a model's pretraining data, as opposed to the bulk of work focusing on detecting data pertinent to a downstream task. To this end, the authors introduce a benchmark dataset -- composed of wikipedia data known to be excluded from current published LLMs -- and a method for predicting if a sample has been seen by the model during pre-training. They compare the performance of their approach with several other known approach, and demonstrate that their approach outperforms the others, on their constructed benchmark dataset. (They also compare direct-examples versus paraphrased examples, though this is in the appendix.) The authors then apply their detection method in two case studies: 1) detecting books known to be included in ChatGPT training data, and 2) detecting data pertinent to downstream tasks. Through these case studies, the authors demonstrate that model size and text length are two important factors in the feasibility of detecting pretraining samples.

**Strengths:**

Although this is not at all my specific area of NLP, I thought the paper was a very nice read. It has a very concrete, and strong motivation for the investigation, and in general is very clearly written, such that I could easily follow the paper, despite this not being my specialty. The authors' proposed method (MIN-K PROB) was very simple and intuitive, and seems to work well across a number of settings (with the WikiMIA benchmark, and also the book and fine-tuning data experiments). I also liked that they included the baseline experiments with paraphrased data. The analysis throughout the paper was also typically very clear, reinforcing the role of model size and text length.

**Weaknesses:**

Because I found the paper mostly very clearly, I don't have much to say in terms of weaknesses :-). Thus, I will use this section to point out the instances that are a bit unclear, or otherwise areas I think should be addressed in some form (e.g. punctuation), though I know this does not really constitute "weaknesses".

Weaknesses:

1. I presume the authors only work with English? That itself is not a weakness, of course, but it should be clearly stated. The "weakness" part of working with only English, is that I think lower-resourced languages may contradict the underlying hypothesis propping up the MIN-K PROB method: That is, because English is such a large percentage of training corpora for multilingual models, I'm not sure how this would work for other languages where the token probabilities given by LLMs are likely to be very low in general. How does this square with the authors' original hypothesis, and newly proposed method?

Minor clarity issues:

2. [S4.1] You say you use "different lengths (32, 64, 128, 256)", but do not specify what -- is this characters? tokens? words? S5.1 says "words" specifically.

3. [S6.2] Similar to above: are your "examples" document level? Sentence level? Or what?

4. [Figure 4] Your figure 4 makes it look as if 7 of the books are contaminated beyond 100%. I can tell from Table 2 that the books in the last bin (100-120) of figure 4 are at 100%, but I would either outright explain that the final bucket corresponds to 100% contamination, or change something with the visualization, because it looks a little funny :-).

4. [S6.2] I find the verbiage around "in-distribution" and "outlier" contaminants to be a bit awkward. If I'm understanding you correctly, Fig5 (a) shows the contaminates that you correctly detect in the continued-pretrained model, (b) shows contaminated unrelated to the continued-pretrained model, but still included in pre-training. Is (a) the "outlier", simply because its not supposed to be included in pre-training data, and thus (b) is the "in-distribution" because this sort of data is presumed to be the type of thing we'd expect in pretraining data?  Maybe reword to "downstream task data" and "control data", or something of the like?

Punctuation:

5. [Appendix] Tables 5 and 6 have no bold numbers.

6. [Page 3, "Challenge 2", second paragraph] Missing a "." after the first sentence.

**Questions:**

1. (From "Weaknesses" Q1) Seeing as English is going to be much more sensitive in terms of word/token probabilities, what about non-English languages, that don't constitute so much of the PLM training data. If we take mT5 as a VERY ROUGH example of the distribution of representation across languages for LLM pre-training data: English represented roughly 5% of training data -- but what about all the languages that represent only a fraction of a percent of training data (e.g. Basque, Hausa, Amharic)? Will your proposed detection method still work for these instances, where the "signal" that each token

2. (From "Weaknesses" Q4) This distinction between "in-distribution" (August 2023 Events Wikipedia) and "outlier" (downstream task datasets that ought nought to be in training) got me thinking -- do you think your approach will struggle with certain domains over others? Again, if we hark back to your original assumption about unseen data having more outliers w.r.t. probability from the LLM: I would expect books and such to have much more complicated (i.e. higher PPL) samples than, say, twitter posts.

---

> ### Author Response · Authors · 2023-11-21
> **Response to Reviewer S4cH**
>
> We thank the reviewer for their constructive comments/feedback. We respond to the reviewer’s comments and questions below.
>
> ---
>
> > I presume the authors only work with English? That itself is not a weakness, of course, but it should be clearly stated. The "weakness" part of working with only English, is that I think lower-resourced languages may contradict the underlying hypothesis propping up the MIN-K PROB method: That is, because English is such a large percentage of training corpora for multilingual models, I'm not sure how this would work for other languages where the token probabilities given by LLMs
>
> Thanks for your feedback! We've revised Section 8, the conclusion section, to include a note that our evaluations were exclusively conducted on English datasets. We also highlight in this section the need for future research to focus on the evaluation of low-resource languages (**Changes in the paper have been highlighted in yellow in the updated PDF**).
>
> While our current focus has been on English, we are optimistic about the applicability of our methods to texts that constitute a minor fraction of the pretraining data, such as non-English corpora. For example, in our dataset contamination experiments, we detected test set contamination, which is only **0.01%** of the pretraining data and our method shows effectiveness in these settings.
>
> The primary challenge we face in extending our evaluation to non-English languages is the complexity and effort required to construct a robust non-English dataset tailored for membership inference attack detection. We welcome suggestions for relevant datasets in non-English languages and would be very happy to test our method on them.
>
> ---
>
> > This distinction between "in-distribution" (August 2023 Events Wikipedia) and "outlier" (downstream task datasets that ought nought to be in training) got me thinking -- do you think your approach will struggle with certain domains over others? Again, if we hark back to your original assumption about unseen data having more outliers w.r.t. probability from the LLM: I would expect books and such to have much more complicated (i.e. higher PPL) samples than, say, twitter posts.
>
> This is a great question! We agree that certain domains may be harder to detect than the others. This detection difficulty difference can stem from unique characteristics inherent to each domain, such as the length and complexity of examples. Our paper presents some preliminary evidence:
>
> * (1) The AUC score for the wiki domain (Table 1) is lower than that for the books domain (Figure 3).
>
> * (2) Outlier data is easier to detect compared with in-distribution data.
>
> Our findings are preliminary, but future research could explore how domain intricacies influence detection efficacy. **We have updated Section 8, our Conclusion, to highlight this as a key area for future research (marked in yellow for easy reference).**
>
> ---
>
> > I find the verbiage around "in-distribution" and "outlier" contaminants to be a bit awkward. If I'm understanding you correctly, Fig5 (a) shows the contaminates that you correctly detect in the continued-pretrained model, (b) shows contaminated unrelated to the continued-pretrained model, but still included in pre-training. Is (a) the "outlier", simply because its not supposed to be included in pre-training data, and thus (b) is the "in-distribution" because this sort of data is presumed to be the type of thing we'd expect in pretraining data? Maybe reword to "downstream task data" and "control data", or something of the like?
>
> Yes. (a) is the outlier because the downstream task data is not supposed to be included in pretraining data, while (b) is the sort of data we'd expect in pretraining. **We have updated the wording in the paper.** Thanks for your suggestion!
>
>
> ---
>
> > You say you use "different lengths (32, 64, 128, 256)", but do not specify what -- is this characters? tokens? words? S5.1 says "words" specifically. Similar to above: are your "examples" document level? Sentence level? Or what?
>
> The length refers to “words” and our examples are sentence-level. Thanks for pointing it out. **We have updated the paper (Section 2.2) to make it clear.**

---

> > ### Comment · Reviewer_S4cH · 2023-11-22
> > **Reply to author response**
> >
> > Thank you for addressing my points here -- I think this has helped the clarity of the paper, for future readers.
> >
> > One small note is that the formatting on your Figure 5 became wonky, after edits. Sub-figure (a) is higher than (b) and (c). This is a minor issue.

---

### Official Review · Reviewer_Z3oB · 2023-11-01

**Soundness:** 3 good
**Presentation:** 3 good
**Contribution:** 3 good
**Rating:** 6
**Confidence:** 3

**Summary:**

This paper presents a method to detect training data in LLMs, based on the assumption of "an unseen example tends to contain a few outlier words with low probabilities, whereas a seen example is less likely to contain words with such low probabilities." Experiments on one wiki dataset verify the advantage of the proposed method over some baselines.

**Strengths:**

1. Training data detection is an important problem to explore.

2. The proposed method does not need any reference model, which is easy to implement.

3. The experimental results show some advantage.

**Weaknesses:**

1. Only one dataset is used for experiments.

2. Please correct me if I am wrong. The proposed method seems to only applicable to LLMs which provide logits of outputs. Can it be applied to SOTA LLMs such as GPT-4 and Claud?

3. Please correct me if I am wrong. The proposed method seems to have difficulty in detecting the texts which are very well written or very badly written.

**Questions:**

See above review

---

> ### Author Response · Authors · 2023-11-20
> **Response to Reviewer Z3oB**
>
> We thank the reviewer for their constructive comments/feedback. We respond to the reviewer’s comments and questions below.
>
> ---
>
> > Only one dataset is used for experiments.
>
> We evaluate our proposed method and baselines on **7 datasets**, including 1 WikiMIA dataset, 5 datasets from our dataset contamination detection experiments (BoolQ, Commonsense QA, IMDB, Truthful QA in Section 5) and 1 copyright book detection experiments (in Section 6). Across 7 datasets, we observe that our proposed method is consistently better than the baselines.
>
> ---
>
> > Please correct me if I am wrong. The proposed method seems to only applicable to LLMs which provide logits of outputs. Can it be applied to SOTA LLMs such as GPT-4 and Claud?
>
> Our method requires output logits from LLMs. OpenAI models such as text-davinci-003 provide such logits. For example, Section 5 shows the effectiveness of our method in detecting copyrighted books from text-davinci-003 (0.88 AUC).
>
> As stated by OpenAI (https://openai.com/blog/new-models-and-developer-products-announced-at-devday), GPT-4 will provide output logits in the next few weeks. We look forward to experimenting with our method on GPT-4.
>
> ---
>
> > Please correct me if I am wrong. The proposed method seems to have difficulty in detecting the texts which are very well written or very badly written.
>
> The proposed method is **effective across any type of text**. including very well-written and very badly written texts. Our method relies on a hypothesis: an unseen example is likely to contain a few outlier words with low probabilities under the LLM, while a seen example is less likely to have words with such low probabilities. This hypothesis is not limited to specific writing styles or quality levels.

---

> ### Author Response · Authors · 2023-11-22
> **Follow-up on our previous response**
>
> Thank you again for your review and feedback! As the discussion period is coming to an end, we want to check in and see if our previous response has addressed your concerns. If you have any follow-up questions or any concerns we haven't addressed yet for a better score, please let us know and we would be happy to answer them.

---

### Meta-Review · Area_Chair_SzE7 · 2023-12-06

**Metareview:**

This paper addresses the task of detecting data that has been used as part of LLM pretraining. The authors introduce a new benchmark, WIKIMIA, that uses data created before and after model training; and a new method, MIN-K PROB, that is based on the hypothesis that “an unseen example is likely to contain a few outlier words with low probabilities under the LLM, while a seen example is less likely to have words with such low probabilities”. The advantages of this approach is that it can be applied without any knowledge of the pretraining corpus or need for a reference model. The authors present experiments and analyses that demonstrate the effectiveness of their approach on a range of datasets. The work is sound and solid, and the paper presents an interesting addition to the research community. I would encourage the authors to address all of the revewers’ concerns in the new version of the paper.

Other comments:
1. Regarding the author response below, I believe the point the reviewer is trying to make is that of ungrammatical text (eg misspellings) and how it might interfere with the hypothesis, as probability of these would be low too. Have “badly writtten texts” been explicitly investigated? It would be good if the authors could expand on this point further.

“The proposed method is effective across any type of text. including very well-written and very badly written texts. Our method relies on a hypothesis: an unseen example is likely to contain a few outlier words with low probabilities under the LLM, while a seen example is less likely to have words with such low probabilities. This hypothesis is not limited to specific writing styles or quality levels.”

2. An issue has been raised by one of the reviewers with respect to why the proposed approach works: “My last question (and the most important one) is that why the proposed method works? Authors have not answered this pivotal question. Why not using the occurrence probability of all the words, why not the occurrence probability of the entire sentence, why not the occurrence probability of the sentences that appear in the pretraining corpus and filtering out the rest. "WHY" the proposed heuristic works?”
The authors should consider adding additional baselines / heuristics to further substantiate their point and effectiveness of their approach.

**Justification For Why Not Higher Score:**

This is an interesting piece of work that would benefit the community; however, it is not of a groundbreaking nature, and, therefore, I believe it is better suited as a poster paper.

**Justification For Why Not Lower Score:**

The work is novel, sound and solid, and the paper presents an interesting addition to the research community.

---

### Decision · Program_Chairs · 2024-01-16

Accept (poster)